# Effect of Different Soil Treatments with Hydrogel on the Performance of Drought-Sensitive and Tolerant Tree Species in a Semi-Arid Region

Ivana Tomášková [1,*], Michal Svatoš [2], Jan Macků [3], Hana Vanická [2], Karolina Resnerová [2], Jaroslav Čepl [1], Jaroslav Holuša [2], Seyed Mohammad Hosseini [3] and Achim Dohrenbusch [4]

[1]  Faculty of Forestry and Wood Sciences, Department of Genetics and Physiology of Forest Trees, Czech University of Life Sciences Prague, Kamýcká 129, Prague 6, CZ-16500 Suchdol, Czechia; cepl@fld.czu.cz

[2]  Faculty of Forestry and Wood Sciences, Department of Forest Protection and Entomology, Czech University of Life Sciences Prague, Kamýcká 129, Prague 6, CZ-16500 Suchdol, Czechia; svatosm@fld.czu.cz (M.S.); vanicka@fld.czu.cz (H.V.); resnerovak@fld.czu.cz (K.R.); holusa@fld.czu.cz (J.H.)

[3]  Faculty of Forestry and Wood Sciences, Department of Forestry Technologies and Construction, Czech University of Life Sciences Prague, Kamýcká 129, Prague 6, CZ-16500 Suchdol, Czechia; macku@fld.czu.cz (J.M.); S_hosseini99@yahoo.com (S.M.H.)

[4]  Faculty of Forest Ecology, George-August University of Göttingen, Wilhelmsplatz 1, D-37073 Göttingen, Germany; adohren@gwdg.de

*  Correspondence: tomaskova@fld.czu.cz; Tel.: +420-224-383-883

**Abstract:** *Research Highlights:* Although a number of forestry studies have found that hydrogel improves tree performance, studies that are located in semi-arid regions and that include a broad spectrum of tree species and the assessment of multiple physiological traits are lacking. *Background and Objectives:* The objective of the current study was to evaluate the effects of hydrogel treatments (with sawdust, organic fertilizer, compost, wheat straw, subsoil, or subsoil with a cobble cover) applied during planting on the survival, growth, and physiological traits of 20 tree species. *Materials and Methods:* In a field experiment (factorial design with seven treatments including a control, 20 species, and ten replicates) in a semi-arid part of Iran, we applied water alone (control) or water with hydrogel and other materials to recently planted samplings. We evaluated tree height, health, osmotic potential, and biochemical properties after 6 months and survival after 12 months. *Results:* Hydrogel treatment (regardless of other material) significantly improved the performance of drought-sensitive but not of drought-tolerant species. *Conclusions:* The benefits of hydrogel treatment are substantial for drought-sensitive species but are insignificant for drought-resistant species.

**Keywords:** drought stress; hydrogel; physiological trait; health status; tree survival

---

## 1. Introduction

Among Middle Eastern countries, Iran faces unprecedented climate change. For example, during the past 40 years, the precipitation decrease was 2.56 mm per year in northwest Iran [1]. An increase in temperature of at least 2 °C is expected in the next decades. This increase in temperature is predicted to be accompanied by a severe decline (~35 %) in precipitation and an anomalous increase in upward longwave radiation, which may be further enhanced by $CO_2$ emissions that are the largest among all Middle Eastern countries [2].

Although Iran is mostly an arid or semi-arid country, part of the country with a temperate climate near the Caspian Sea is becoming semi-arid because of the climate change [3]. Hyperspectral images and vegetation indexes indicate that climate change and human population dynamics have reduced the

forest area in Iran by about 43% from 1972 to 2009 [4,5]. Reforestation is required to increase landscape water-holding capacity and also to increase biomass accumulation and, therefore, carbon storage [6].

Many research articles have evaluated the positive effects of hydrogel on selected forest tree species [7–9]. Hydrogel is a synthetic polyacrylamide with a large capacity for water retention and storage. Application of hydrogel can counter conditions of low precipitation, high evapotranspiration demand, and low soil water-holding capacity.

Trees and other plants grown in hydrogel-treated soil showed improved physiological and morphological traits and increased survival, water-use efficiency, and dry matter production [10,11]. By forming a flexible envelope, hydrogel mimics the effects of mucilage naturally exuded by roots in order to maintain ion and water exchange processes between the root system and the rhizosphere. Root exudates containing amino acids and saccharides positively influence the soil microorganisms and therefore increase the availability of organic and inorganic nutrients to roots [12]. The leaves of hydrogel-treated beech and eucalyptus trees contained higher contents of nutrients (N, K, and Mg) than the leaves of untreated trees [13,14]. Hydrogel-treated plants generally require 20% less fertilizer than untreated plants [13]). By increasing the uptake of $Ca^+$, $Na^+$, and $Cl^-$, hydrogel may help plants survive the combination of drought and salt stress [15].

Positive effects of hydrogel have been reported by many studies, but several studies have reported non-significant effects of hydrogel on plant growth [13]. Germination seems to be the only trait negatively influenced by hydrogel application [8,12]. Hydrogel might suppress germination because the polymer is able to withhold large amounts of water when excessive irrigation is applied [10]. When used alone, hydrogel may also harm seedlings by reducing root aeration [12,16]. Therefore, the application of an organic substrate (vermiculite, bark, etc.) along with hydrogel is highly recommended to increase soil porosity and thus facilitate root and rhizosphere gas exchange. For the same reason, hydrogel may also be less effective in clay or loamy soils than in sandy soils [17].

Although many studies have documented positive effects of hydrogel on plant growth, a comprehensive study with different mixture treatments with hydrogel content and with several tree species in a semi-arid ecosystem is lacking. The goal of the current research was to conduct such a study with a focus on the morphological and physiological traits of the studied species. In a replicated field experiment, we compared the effects of hydrogel treatments with a non-treated control on the performance of 20 common tree species that differ in their drought resistance.

## 2. Materials and Methods

The research was conducted at 15 km east of Tehran in 2008 (35.819 N, 51.511 E; 1857 m a.s.l.). The average slope was 8%, and slopes were oriented to the southeast. The average annual precipitation was 208 mm, and most precipitation occurred in the winter. Mean annual rainfall ranged from 51 to 1835 mm [18]. The soil was a sandy loam with a pH of 7.5 and an extremely low N content (0.02% on a mass base).

The experiment used 2-year-old saplings that had been grown in a plantation in Karaj, which is 50 km from Tehran. The experiment occupied 2500 $m^2$ and was protected against livestock and rodents by a 1.5-m-high metal mesh fence. During the spring of 2008, the saplings were planted 2.0 m from each other at the study site. The following seven treatments were applied immediately after planting:

(1)   CO (control)
(2)   OM (hydrogel and sawdust)
(3)   HM (hydrogel and organic fertilizer)
(4)   HC (hydrogel and compost)
(5)   HK (hydrogel and wheat straw)
(6)   HS (hydrogel and subsoil)
(7)   HP (hydrogel and subsoil with a cobbles cover)

The seven treatments were applied in a factorial design to 20 tree species (Table 1) with 10 replicate saplings for each combination of treatment and tree species, giving a total of 1400 plants (7 treatments × 20 species × 10 replicates).

A 1250-g quantity of the hygroscopic polymer (trade name STOCKOSORB® 300; Evonik Nutrition & Care GmbH, Essen, Germany) was added to 100 L of water. After it congealed, 2 L of the mixture (plus added materials) was applied to the soil in the root zone of each sapling. The amount of added material (wheat straw, compost, subsoil, sawdust) was approx. 1/3 of the total volume that was added to the roots. The fertilizer in the form of capsules (10 g, with N, P, K nutrients) was administered to the roots together with the hydrogel. The control plots received no fertilizer.

The following parameters were evaluated 6 months after the saplings were planted and treated: height, health (on a scale from 1 to 4, where 1 was a dead plant and 4 was a vigorously growing plant), and leaf osmotic potential at 4 pm (measured during 1 week in 2008). In total, six leaves per treatment and tree species were measured. The vapor-pressure osmometer Wescor Model 5500 (Wescor, UT, USA) was used for measuring osmotic potential. To estimate drought vulnerability, previous reports of $P_{50}$ values for each species were used. $P_{50}$ is the leaf water potential at which 50% of hydraulic conductivity is lost (Table 1). Survival was assessed 1 year after planting.

**Table 1.** Origin and vulnerability to drought stress of studied trees (vulnerability is indicated by $P_{50}$ values reported by the following authors: [19–23] reported as superscripts in the last column).

| Tree Species | Origin | Vulnerability to Drought Stress ($P_{50}$, MPa) |
|---|---|---|
| *Acer negundo* L. | N America | −1.7 [19] |
| *Ailanthus altissima* (Mill.) Swingle | E Asia | −1.2 [19] |
| *Azadirachta indica* A. Juss. | SE Asia | unknown |
| *Berberis vulgaris* L. | Europe, W Asia | −5.7 [19] |
| *Celtis caucasica* Willd. | SE Europe, Himalayas | from −0.8 to −1.5 (*Celtis* spp.) [19] |
| *Cercis siliquastrum* L. | W Asia, Mediterranean | −1.8 [19] |
| *Cupressus arizonica* Greene | N America | −11.0 (*Cupressus* spp.) [19] |
| *Elaeagnus angustifolia* L. | E Europe, Asia | unknown |
| *Erythrostemon gilliesii* (Hook.) Klotzsch | S America | from −2.1 to −2.5 (*Erythrostemon* spp.) [19] |
| *Fraxinus excelsior* L. | Europe | −2.8 [19] |
| *Juglans nigra* L. | N America | −2.0 [19] |
| *Juniperus sp.* | Holarctic region | −9.8 [19] |
| *Morus alba* L. | China | −0.2 [19] |
| *Olea europea* L. | S Europe, N Africa, W Asia | −7.2 [19] |
| *Pinus brutia* subsp. *eldarica* Tenore | Middle East, Russia | −3.1 [19] |
| *Pinus nigra* J. F. Arnold | S Europe | from −2.8 to −3.8 [20] |
| *Platanus orientalis* L. | SE Europe, SW Asia | −1.8 (*Platanus* hybrids) [21] |
| *Platycladus orientalis* (L.) Franco | Asia, Russia | −3.6 (*Platycladus* spp.) [19] |
| *Populus nigra* L. | Europe, Asia | −2.9 [19] |
| *Robinia pseudoacacia* L. | N America | from −0.5 to −0.9 [22,23] |

Biochemical analyses of leaf material were performed 6 months after planting. Biochemical parameters were analyzed in control and HS (hydrogel with subsoil) treatments only. The methodology is briefly described in the following paragraphs.

## 2.1. Photosynthetic Pigments

Photosynthetic pigments were extracted from an assimilation apparatus in acetone. The extract was passed through filter paper and centrifuged. The pigment content was calculated using the following equations [24]:

$$c(chl\ a) = 12.21 \times A_{663} - 2.81 \times A_{646} \left[ \text{mg L}^{-1} \right] \qquad (1)$$

$$c(chl\ b) = 20.13 \times A_{646} - 5.03 \times A_{663} \left[ \text{mg L}^{-1} \right] \qquad (2)$$

$$c\ (car) = 1000 \times A_{470} - 3.27 \times c(chl\ a) - 104 \times c(chl\ b))/198 \left[mg\ L^{-1}\right] \tag{3}$$

Concentration of chlorophyll a [$c(chl\ a)$] and b [$c(chl\ b)$] and carotenoids $c(car)$ is calculated from the pigment absorbance at different wavelengths $(A_{663,\ 646,\ 470})$ measured spectrophotometrically (DR 3900, Hach Company, Loveland, CO, USA).

### 2.2. Rubisco

The activity of ribulose-l,5-bisphosphate carboxylase oxygenase (Rubisco) was measured with a spectrophotometer according to Barta et al. 2011 [25].

### 2.3. Nitrogen

The N content in dried (48 h, 80 °C) and weighed leaves or needles was measured with the automatic elemental analyzer (CNS-2000, LECO Corp., St. Joseph, MI, USA). Commercial standards (sulfamethazine and alfalfa) were used for calibration.

### 2.4. Electrical Conductivity

The electrical conductivity of stem samples was measured with a Milwaukee Mi180 multimeter after the samples were cooled to 5 °C, warmed to 20 °C for 1 h, and then autoclaved. Electrical conductivity was measured as the ratio of values before and after autoclaving.

### 2.5. Prolin

Prolin in an assimilation apparatus was estimated with the spectrophotometer (DR 3900, HACH Company, Loveland, MI, USA) based on the reaction between amino acid ninhydrin (2,2-dihydroxyindane-1,3-dione) [26].

### 2.6. Abscisic Acid

ABA was measured by GS-MS according to Okamoto et al. [27].

### 2.7. Total Protein

Protein concentration in assimilation was determined according to Bradford using bovine serum albumin as the standard [28].

### 2.8. Zinc

Zn content in assimilation was measured with an atomic absorption spectrophotometer (AA-670, Shimadzu, Columbia, MD, USA) according to the methods of Diatloff and Rengel (2001) [29].

### 2.9. Relative Water Content

For determination of RWC, the weight of leaves or needles was measured in their fresh state (FW), after immersion for 3 h in de-ionized water (TW), and after drying in an oven (DW). RWC was calculated according to the following equation:

$$RWC\ (\%) = [(FW\text{-}DW)/(TW\text{-}DW)] \times 100 \tag{4}$$

In order to make the statistical results clearer, the findings were expressed for two treatments only: control one and all treatments with hydrogel addition were embodied into one hydrogel treatment (although no pure hydrogel treatment was used). All treatments with hydrogel addition (e.g., OM, HM, HC, HK, HS, HP) were joined together into one group. Finally, two groups were compared—control and hydrogel treatments. Nevertheless, the detailed results of statistical differences among all tree species and all seven treatments are part of Appendix A.

The selected trees included 15 broadleaf and 5 coniferous species that are common in Western Asia (Table 1). For better visibility of key findings, we divided tree species into two groups. Drought-tolerant (TOL) and drought sensitive (SEN) tree species were selected on the basis of $P_{50}$ and differences in biochemical parameters. TOL species comprise *Olea*, *Eleagnus,* and both *Pinus* species, the rest of the investigated tree species was part of the SEN group.

Statistical analysis was conducted using R software environment, version 3.6.1 (R Core Team, 2019). Plots were made using packages ggplot2 [30] and fmsb [31].

Prior to statistical analyses, data were checked for normality and log-transformed when appropriate.

To assess differences among treatments and species combinations, a generalized least squares model was applied (gls function from nmle package) [32], which allowed to model specific variance structures. Possible variance structures were fitted, and candidate models were compared by Akaike's Information Criterion (AIC).

The model with the lowest AIC value was used. The normality of the distribution of the residuals was assessed using QQ-plots.

The model can be written in a form:

$$\log(\text{height})_{ijkl} = \text{intercept} + \text{species}_j + \text{treatment}_k + \text{species}_j\text{:treatment}_k + \varepsilon_{il} \tag{5}$$

where $\log(\text{height})_{ijkl}$ is log-transformed height increase of ith observation, for jth species, kth treatment and lth site; species is categorical variable with two levels (sensitive and tolerant) and treatment is also categorical variable with two levels (control and hydrogel); $\varepsilon_{ij}$ as an error term of ith observation at lth site. Variance of errors was specified as varIdent(~1|site), allowing estimation of different variances at each of 10 sites and varExp(form = ~height) combined together with varComp function. Residuals are normally distributed with mean 0 and $\text{var}(\varepsilon_{il}) = \sigma_l^2 \times e^{2\delta \times yi}$:

$$\varepsilon_{il} \sim N\left(0, \sigma_l^2 \times e^{2\delta \times height\ i}\right) \tag{6}$$

where δ is the estimated parameter.

Post-hoc pairwise comparisons were performed by the Tukey HSD method implemented in emmeans function from emmeans package. As we worked with log-transformed data, reported values represent back transformed means.

For survival data, prop.test function from base R was used to perform Pearson's chi-squared test statistic to evaluate the significance of differences among hydrogel treatments. Subsequently, post-hoc pairwise comparison was performed using *p*-values adjusted by Holm's method for correction for multiple testing as implemented in pairwise.prop.test function from base R. Detailed information is part of Tables A1–A3.

## 3. Results

Relative to the control, each of the treatments with hydrogel significantly increased the survival rate and positively influenced the height increases and health status of all tree species (Figures 1 and 2).

Both drought-tolerant (TOL) and drought-sensitive (SEN) were in very poor condition (no excellent health status) without hydrogel. Survival of all species was 37% in control and 81% in all hydrogel treatments together. The application of hydrogel together with subsoil and cobbles on the surface was especially beneficial for *Eleagnus*, *Robinia*, and *Juniperus* species. The increase in height for the treatment with cobbles was 35%–45% greater than for the other hydrogel treatments (Table A4). Health status was highest for *Pinus nigra*, *Olea*, and *Erythrostemon* species when they were treated with hydrogel and organic fertilizer or compost (*Pinus* and *Olea*), or with hydrogel and sawdust or subsoil (*Olea* and *Erythrostemon*). For *Erythrostemon*, a health status of four (i.e., excellent) was recorded for the hydrogel treatment with sawdust, and 70% of all trees were considered to be in perfect shape. For the untreated control, no *Erythrostemon* tree had a health status of four, and 70% had a health status less than three.

No matter what kind of hydrogel treatment was used *Olea*, *Cercis*, *Celtis*, and *Erythrostemon* species showed the highest rate of survival compared to other tree species with any hydrogel treatment. On the other hand, survival of *Platanus*, *Acer*, and *Juglans* was similar among untreated control trees and those treated with hydrogel (Table A4).

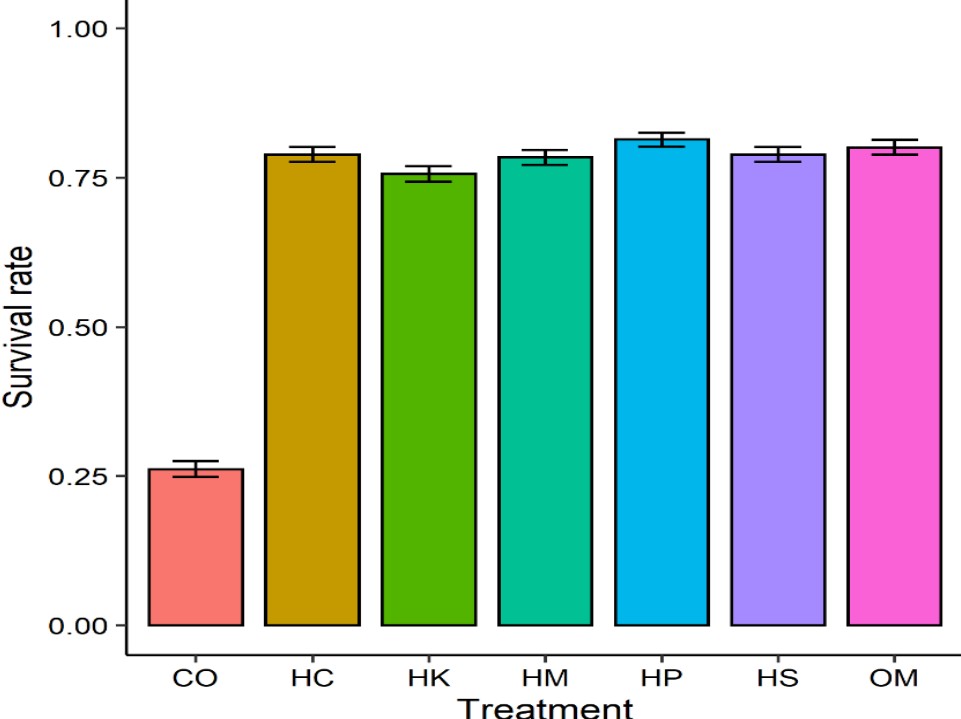

**Figure 1.** Different survival rates for all species at all sites over the studied period under different hydrogel treatments and control. According to multiple proportion test, CO is significantly different than any hydrogel treatment, whereas different hydrogel treatments do not differ significantly. N = 1080 for each treatment.

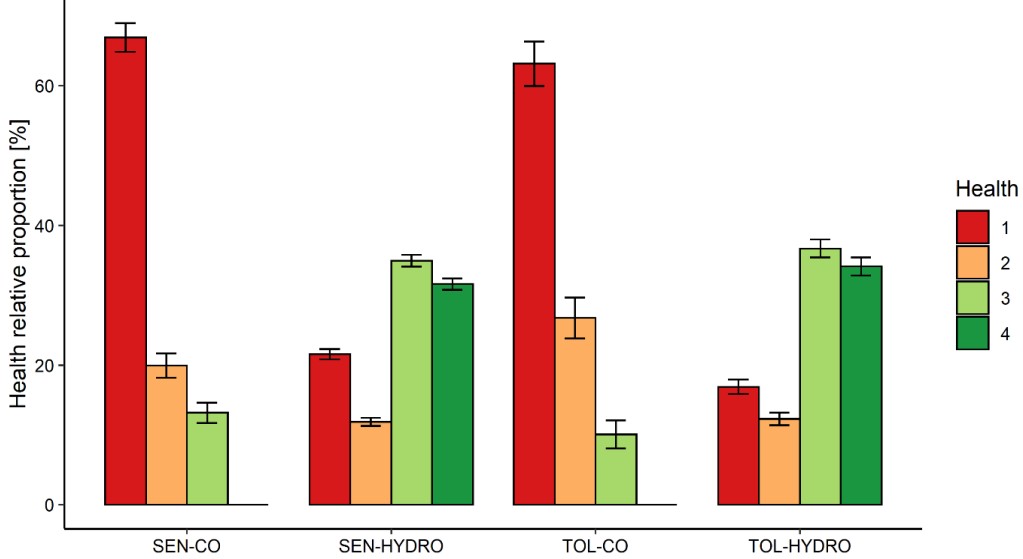

**Figure 2.** Relative proportion of health category within drought-tolerant (TOL) and drought-sensitive (SEN) species with (HYDRO) or without (CO) hydrogel treatment (all treatments with hydrogel addition together). Health = 4 denoting healthy plant and health = 1 denoting dying plant. $N_{TOL-CO} = 228$; $N_{TOL-HYDRO} = 1365$; $N_{SEN-CO} = 532$; $N_{SEN-HYDRO} = 3192$; SE indicated as error bars.

The key biochemical traits were influenced in the HS treatment (hydrogel with subsoil) compared to control one (Figure 3 and Table A5). Nevertheless, no statistically significant differences in biochemical traits between control and hydrogel treatments were found.

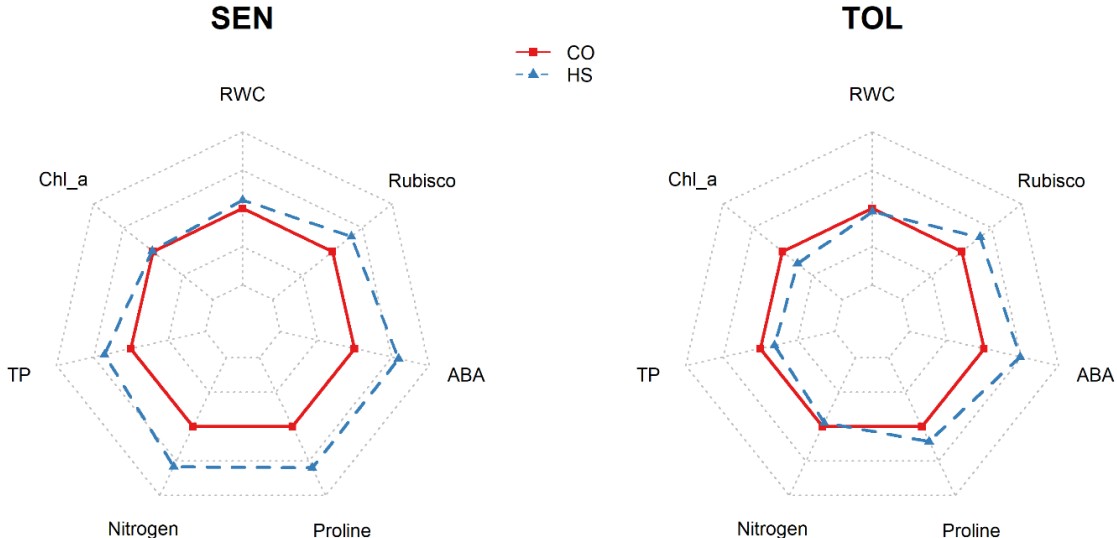

**Figure 3.** Spider-plot of biochemical variables in drought-tolerant (TOL) and drought-sensitive species (SEN) with (HS = hydrogel + subsoil) or without (CO) hydrogel. Values are displayed relative to respective CO treatment, and each concentric segment of a web denotes a 10% difference in the given parameter.

## 4. Discussion

Hydrogel application has been reported to enhance the aboveground growth of the plant, support survival, and significantly improve the health status [10,11,14,16]. Moreover, our findings indicate that the magnitude of the response to hydrogel strongly depends on the drought-sensitivity or drought-tolerance of the tree species. In response to hydrogel treatment, the increases in leaf relative water content (RWC) and leaf dry mass (LDM) were expected. Both of these traits were previously recognized as common effects of hydrogel treatment [11]. Similar to the results of M'barki [33] hydrogel increased growth, dry weight, RWC because of stronger uptake of the micro- and macronutrients, especially nitrogen and potassium [34]. With a higher accumulation of nutrients, the increased amounts of total proteins, photosynthetic pigments and Rubisco could be explained [35]. Similarly, a higher quantum yield (Fv/Fm) or higher photosynthetic rate [8] is usually reported for hydrogel-treated plants. One of the main benefits of hydrogel treatment is an increase in prolin. In both the SEN and TOL group, higher prolin concentration was recorded. From this point of view, no correlation with osmotic potential or RWC is surprising because the osmolytic properties of prolin usually explain higher water potential in hydrogel-treated plants [36]. In addition to the osmoprotective function, it acts as a metal chelator or antioxidant [37]. Moreover, the positive effect of hydrogel treatment was proved by lower concentration of malonaldehyde (MDA) or higher activity of superoxide dismutase (SOD) [38]. Higher concentration of MDA stands for membrane destruction, and the presence of SOD suggests the presence of reactive oxygen species—both signs of drought stress that hydrogel treatment can alleviate.

Our findings imply that hydrogel efficacy depends on tree species susceptibility to environmental stress. Based on our results, the hydrogel application is more beneficial for drought-sensitive tree species compared to drought-tolerant ones. Drought-sensitive trees typically inhabit deep soils that contain sufficient organic matter and with sufficient supplies of water and nutrients. Their $P_{50}$ values do not exceed $-2.0$ MPa (see Table 1). *Cupressus*, in particular, is considered as one of the most drought-tolerant tree species [19]. Moreover, hydrogel should be applied with additional substances as straw or subsoil that can increase soil porosity and, therefore, oxygen supply to the roots. In this respect,

an expanded form of vermiculite can be effective [16]. Added substrate porosity seems to be crucial for any hydrogel amendment treatments as poor physiological tree growth may be related to the reduced root aeration [12]. The different mixtures of various substances with hydrogel probably explains the differences among hydrogel treatments in their effects on morphological traits and why hydrogel with subsoil was often the best treatment for any particular morphological trait. The treatment with hydrogel, subsoil, and cobbles was also favorable for height increase. This confirms that the traditional practice of cobbles placement in the Anasazi Rio Grande landscape in the USA improves local water management by trapping seasonal runoff and reducing evaporation [39].

## 5. Conclusions

The current study confirms the positive effects of hydrogel treatments on the tree growth and survival. Nevertheless, the magnitude of the response to the hydrogel treatment depends on the drought vulnerability with a positive effect on drought-sensitive species. The addition of hydrogel during planting is a low-cost and effective way of preserving all tree species in the region with upcoming climate change. As no special equipment or machinery is necessary for hydrogel application, this method seems to be a very promising tool for climate change mitigation in forestry planting. Moreover, the hydrogel treatment could secure the timber production in countries currently facing drought or elevated temperatures.

**Author Contributions:** I.T. wrote the manuscript; J.Č. performed the statistical analyses and data visualizations; S.M.H. and A.D. managed the research and the collection of material and took part in finalizing the manuscript; M.S., H.V., J.M., K.R., and J.H. supervised the literature research and methods, and edited the manuscript. All authors have read and agreed to the published version of the manuscript.

**Funding:** This research was funded by the EVA4.0—No. CZ.02.1.01/0.0/0.0/16_019/0000803 financed by OP RDE and by the National Agency for Agricultural Research within the framework of projects under grant No. QK1910347 administered by the Ministry of Agriculture of the Czech Republic.

**Acknowledgments:** The authors thank Bruce Jaffee (USA) for linguistic and editorial improvements.

**Conflicts of Interest:** The authors declare no conflict of interest.

## Appendix A

**Table A1.** Estimated marginal means (emmeans) and confidence interval for species and treatment combinations as the outcome of the emmeans function. Results are given on the log (not the response) scale. Confidence level used: 0.95.

| Species | Treatment | Emmean | SE | df | Lower.CL | Upper.CL |
|---------|-----------|--------|--------|------|----------|----------|
| SEN | CO | 2.23 | 0.04505 | 3619 | 2.14 | 2.32 |
| TOL | CO | 1.89 | 0.07821 | 2979 | 1.74 | 2.04 |
| SEN | HYDRO | 2.38 | 0.00983 | 4345 | 2.36 | 2.40 |
| TOL | HYDRO | 2.12 | 0.01857 | 3930 | 2.08 | 2.15 |

**Table A2.** Pairwise comparison of respective species and treatment combinations. Results are given on the log (not the response) scale. *P*-value adjustment: Tukey method for comparing a family of 4 estimates.

| Contrast | Estimate | SE | df | *t*-Ratio | *p*-Value |
|----------|----------|--------|------|-----------|-----------|
| CO_SEN—CO_TOL | 0.339 | 0.0903 | 3170 | 3.759 | 0.0010 |
| CO_SEN—HYDRO_SEN | −0.150 | 0.0461 | 3735 | −3.254 | 0.0063 |
| CO_SEN—HYDRO_TOL | 0.111 | 0.0487 | 3858 | 2.278 | 0.1034 |
| CO_TOL—HYDRO_SEN | −0.489 | 0.0788 | 3020 | −6.208 | <0.0001 |
| CO_TOL—HYDRO_TOL | −0.228 | 0.0804 | 3081 | −2.840 | 0.0235 |
| HYDRO_SEN—HYDRO_TOL | 0.261 | 0.0210 | 4056 | 12.423 | <0.0001 |

**Table A3.** Multiple comparisons of survival ratios on different hydrogel treatments. Before multiple comparisons, 7-sample test for equality of proportions was run with $\chi 2 = 554.78$, df = 6 and *p*-value < $2.2 \times 10^{-16}$.

| Treatment | CO | HC | HK | HM | HP | HS |
|:---:|:---:|:---:|:---:|:---:|:---:|:---:|
| HC | $<2 \times 10^{-16}$ | - | - | - | - | - |
| HK | $<2 \times 10^{-16}$ | 1.00 | - | - | - | - |
| HM | $<2 \times 10^{-16}$ | 1.00 | 1.00 | - | - | - |
| HP | $<2 \times 10^{-16}$ | 1.00 | 0.16 | 1.00 | - | - |
| HS | $<2 \times 10^{-16}$ | 1.00 | 1.00 | 1.00 | 1.00 | - |
| OM | $<2 \times 10^{-16}$ | 1.00 | 0.74 | 1.00 | 1.00 | 1.00 |

Table A4. Effects of the seven treatments on the increase in height, health status, and survival for each of the 20 tree species. Tree species are indicated by the first four or five letters of the genus (see Table 1 for full names). Treatments were CO (control), OM (hydrogel and sawdust), HM (hydrogel and organic fertilizer), HC (hydrogel and compost), HK (hydrogel and wheat straw), HS (hydrogel and subsoil), and HP (hydrogel and subsoil with a cobbles cover). Osmotic potential is in negative values. Values are means ± SD.

| Treatment | Species | Health | Height Increase (cm) | Survival (%) | Osmotic Potential (bar) | Species | Health | Height Increase (cm) | Survival (%) | Osmotic Potential (bar) |
|---|---|---|---|---|---|---|---|---|---|---|
| CO | Acer | 1.3 ± 0.5 | 14.9 ± 7.5 | 23.7 ± 43.1 | 15.3 ± 7.9 | Jugla | 1.3 ± 0.6 | 9.0 ± 3.6 | 21.1 ± 41.3 | 14.0 ± 7.3 |
| HC | Acer | 2.6 ± 1.1 | 26.1 ± 17.5 | 71.1 ± 46.0 | 7.4 ± 6.2 | Jugla | 2.4 ± 1.2 | 15.9 ± 9.5 | 65.8 ± 48.1 | 11.1 ± 2.5 |
| HK | Acer | 2.5 ± 1.0 | 22.4 ± 15.2 | 73.7 ± 44.6 | 13.6 ± 5.6 | Jugla | 2.4 ± 1.1 | 17.0 ± 9.2 | 68.4 ± 47.1 | 10.3 ± 5.3 |
| HM | Acer | 2.3 ± 1.3 | 26.0 ± 21.9 | 57.9 ± 50.0 | 12.8 ± 6.0 | Jugla | 2.3 ± 1.4 | 14.4 ± 11.1 | 50.0 ± 50.7 | 9.9 ± 5.4 |
| HP | Acer | 2.4 ± 1.3 | 18.9 ± 8.9 | 57.9 ± 50.0 | 10.7 ± 5.7 | Jugla | 2.0 ± 1.0 | 14.4 ± 10.0 | 57.9 ± 50.0 | 11.3 ± 6.0 |
| HS | Acer | 2.3 ± 1.2 | 23.0 ± 13.9 | 63.2 ± 48.9 | 14.8 ± 4.5 | Jugla | 2.4 ± 1.1 | 15.3 ± 10.0 | 63.2 ± 48.9 | 13.3 ± 3.7 |
| OM | Acer | 2.5 ± 1.3 | 26.0 ± 21.6 | 60.5 ± 49.5 | 13.7 ± 6.9 | Jugla | 2.4 ± 1.2 | 13.7 ± 9.3 | 68.4 ± 47.1 | 10.8 ± 6.3 |
| CO | Ailan | 1.6 ± 0.8 | 25.8 ± 20.8 | 42.1 ± 50.0 | 20.6 ± 1.7 | Junip | 1.5 ± 0.7 | 6.9 ± 2.8 | 36.8 ± 48.9 | 22.0 ± 0.0 |
| HC | Ailan | 2.6 ± 1.3 | 22.3 ± 9.3 | 78.9 ± 41.3 | 18.3 ± 2.6 | Junip | 3.1 ± 1.0 | 8.8 ± 3.7 | 86.8 ± 34.3 | 21.3 ± 1.0 |
| HK | Ailan | 2.9 ± 1.2 | 36.5 ± 28.2 | 73.7 ± 44.6 | 19.9 ± 1.9 | Junip | 2.7 ± 1.1 | 11.9 ± 6.6 | 73.7 ± 44.6 | 21.4 ± 0.9 |
| HM | Ailan | 2.6 ± 1.2 | 19.0 ± 6.6 | 71.1 ± 46.0 | 19.3 ± 3.4 | Junip | 2.9 ± 1.0 | 8.2 ± 3.8 | 89.5 ± 31.1 | 22.0 ± 0.0 |
| HP | Ailan | 2.8 ± 1.1 | 24.6 ± 12.4 | 81.6 ± 39.3 | 18.8 ± 2.4 | Junip | 2.9 ± 1.0 | 13.2 ± 8.2 | 84.2 ± 37.0 | 18.0 ± 8.9 |
| HS | Ailan | 2.7 ± 1.2 | 21.5 ± 9.6 | 76.3 ± 43.1 | 18.5 ± 2.3 | Junip | 2.6 ± 1.2 | 12.1 ± 6.0 | 71.1 ± 46.0 | 21.6 ± 1.0 |
| OM | Ailan | 2.8 ± 1.1 | 22.2 ± 11.3 | 81.6 ± 39.3 | 18.3 ± 2.6 | Junip | 3.1 ± 1.1 | 9.4 ± 5.5 | 89.5 ± 31.1 | 21.3 ± 1.6 |
| CO | Azadi | 1.6 ± 0.9 | 24.3 ± 15.4 | 35.3 ± 48.5 | 21.3 ± 1.0 | Morus | 1.4 ± 0.7 | 18.9 ± 8.6 | 31.6 ± 47.1 | 19.5 ± 2.4 |
| HC | Azadi | 3.1 ± 1.0 | 17.8 ± 8.6 | 88.2 ± 32.7 | 21.5 ± 1.2 | Morus | 2.9 ± 1.1 | 20.9 ± 10.6 | 81.6 ± 39.3 | 18.4 ± 1.4 |
| HK | Azadi | 2.6 ± 1.2 | 17.9 ± 8.9 | 76.5 ± 43.1 | 19.8 ± 1.8 | Morus | 3.1 ± 0.9 | 26.5 ± 16.7 | 89.5 ± 31.1 | 16.4 ± 3.1 |
| HM | Azadi | 3.2 ± 1.1 | 27.4 ± 24.3 | 85.3 ± 35.9 | 22.0 ± 0.0 | Morus | 3.2 ± 0.8 | 23.5 ± 11.7 | 94.7 ± 22.6 | 20.3 ± 3.6 |
| HP | Azadi | 3.1 ± 1.0 | 23.8 ± 11.1 | 85.3 ± 35.9 | 19.1 ± 2.8 | Morus | 3.1 ± 1.0 | 28.7 ± 22.5 | 89.5 ± 31.1 | 17.8 ± 3.3 |
| HS | Azadi | 2.9 ± 1.1 | 22.0 ± 12.0 | 85.3 ± 35.9 | 18.9 ± 3.4 | Morus | 3.1 ± 1.1 | 25.3 ± 14.1 | 86.8 ± 34.3 | 18.1 ± 3.6 |
| OM | Azadi | 3.0 ± 1.1 | 25.1 ± 17.4 | 82.4 ± 38.7 | 20.0 ± 1.8 | Morus | 3.1 ± 1.2 | 25.2 ± 11.8 | 81.6 ± 39.3 | 18.2 ± 3.0 |
| CO | Berbe | 1.4 ± 0.5 | 21.8 ± 22.7 | 36.8 ± 48.9 | 21.5 ± 1.2 | Olea | 1.8 ± 0.9 | 9.9 ± 9.4 | 52.6 ± 50.6 | 22.0 ± 0.0 |
| HC | Berbe | 2.4 ± 1.2 | 21.1 ± 16.5 | 68.4 ± 47.1 | 21.0 ± 1.5 | Olea | 3.4 ± 0.8 | 11.3 ± 10.7 | 97.4 ± 16.2 | 20.0 ± 2.4 |
| HK | Berbe | 2.4 ± 1.1 | 18.4 ± 14.2 | 71.1 ± 46.0 | 17.8 ± 8.8 | Olea | 3.1 ± 0.9 | 14.4 ± 10.3 | 94.7 ± 22.6 | 21.7 ± 0.5 |
| HM | Berbe | 2.4 ± 1.1 | 22.5 ± 17.9 | 76.3 ± 43.1 | 21.3 ± 1.2 | Olea | 3.4 ± 0.7 | 15.8 ± 13.7 | 100.0 ± 0.0 | 20.3 ± 1.5 |
| HP | Berbe | 2.4 ± 1.1 | 20.3 ± 16.5 | 68.4 ± 47.1 | 21.5 ± 0.8 | Olea | 3.1 ± 0.8 | 13.5 ± 8.1 | 94.7 ± 22.6 | 20.3 ± 2.3 |
| HS | Berbe | 2.3 ± 1.2 | 25.1 ± 21.9 | 71.1 ± 46.0 | 21.8 ± 0.4 | Olea | 3.1 ± 0.9 | 15.1 ± 13.3 | 92.1 ± 27.3 | 21.2 ± 1.0 |
| OM | Berbe | 2.9 ± 1.1 | 24.5 ± 21.0 | 78.9 ± 41.3 | 21.0 ± 1.5 | Olea | 3.3 ± 0.8 | 15.5 ± 12.8 | 97.4 ± 16.2 | 19.2 ± 1.3 |
| CO | Celti | 1.7 ± 0.7 | 15.0 ± 7.7 | 52.6 ± 50.6 | 21.3 ± 1.2 | Peld | 1.4 ± 0.7 | 9.2 ± 5.6 | 31.6 ± 47.1 | 18.0 ± 3.4 |

**Table A4.** *Cont.*

| Treatment | Species | Health | Height Increase (cm) | Survival (%) | Osmotic Potential (bar) | Species | Health | Height Increase (cm) | Survival (%) | Osmotic Potential (bar) |
|---|---|---|---|---|---|---|---|---|---|---|
| HC | Celti | 3.1 ± 0.6 | 18.6 ± 8.8 | 100.0 ± 0.0 | 19.3 ± 2.4 | Peld | 2.7 ± 1.0 | 9.5 ± 4.2 | 81.6 ± 39.3 | 19.7 ± 2.3 |
| HK | Celti | 3.0 ± 0.8 | 16.5 ± 10.1 | 94.7 ± 22.6 | 17.9 ± 4.5 | Peld | 2.5 ± 1.2 | 1.5 ± 4.8 | 68.4 ± 47.1 | 17.0 ± 4.5 |
| HM | Celti | 3.1 ± 0.7 | 19.5 ± 10.7 | 94.7 ± 22.6 | 21.1 ± 1.5 | Peld | 2.6 ± 1.2 | 10.0 ± 4.4 | 65.8 ± 48.1 | 16.2 ± 8.6 |
| HP | Celti | 3.2 ± 0.8 | 16.2 ± 10.6 | 97.4 ± 16.2 | 19.4 ± 2.2 | Peld | 2.4 ± 1.2 | 8.9 ± 3.9 | 68.4 ± 47.1 | 17.5 ± 1.6 |
| HS | Celti | 3.1 ± 0.7 | 17.0 ± 9.7 | 97.4 ± 16.2 | 20.0 ± 1.8 | Peld | 2.5 ± 1.2 | 13.2 ± 7.9 | 63.2 ± 48.9 | 20.0 ± 3.1 |
| OM | Celti | 3.2 ± 0.7 | 16.4 ± 7.0 | 97.4 ± 16.2 | 17.8 ± 4.9 | Peld | 2.6 ± 1.1 | 11.5 ± 6.6 | 73.7 ± 44.6 | 15.8 ± 2.6 |
| CO | Cerci | 1.6 ± 0.7 | 7.3 ± 3.5 | 50.0 ± 50.7 | 22.0 ± 0.0 | Plata | 1.0 ± 0.2 | 5.0 ± 0.8 | 2.6 ± 16.2 | 12.8 ± 10.1 |
| HC | Cerci | 3.3 ± 0.7 | 9.8 ± 6.8 | 100.0 ± 0.0 | 21.3 ± 1.0 | Plata | 2.1 ± 1.3 | 18.4 ± 12.1 | 47.4 ± 50.6 | 14.5 ± 2.5 |
| HK | Cerci | 3.2 ± 0.7 | 11.8 ± 7.6 | 97.4 ± 16.2 | 20.3 ± 2.6 | Plata | 1.8 ± 1.1 | 16.3 ± 11.1 | 39.5 ± 49.5 | 15.3 ± 3.4 |
| HM | Cerci | 3.3 ± 0.8 | 8.8 ± 5.4 | 94.7 ± 22.6 | 21.8 ± 0.4 | Plata | 1.8 ± 1.1 | 20.2 ± 11.3 | 36.8 ± 48.9 | 13.2 ± 2.8 |
| HP | Cerci | 3.1 ± 0.6 | 7.2 ± 3.9 | 94.7 ± 22.6 | 20.5 ± 2.0 | Plata | 1.6 ± 1.1 | 13,9 ± 6.0 | 31.6 ± 47.1 | 13.7 ± 2.7 |
| HS | Cerci | 3.1 ± 0.7 | 10.4 ± 7.5 | 97.4 ± 16.2 | 22.0 ± 0.0 | Plata | 1.7 ± 1.1 | 17.3 ± 5.0 | 31.6 ± 47.1 | 14.8 ± 3.1 |
| OM | Cerci | 3.2 ± 0.8 | 11.8 ± 10.9 | 92.1 ± 27.3 | 21.5 ± 1.2 | Plata | 1.7 ± 1.0 | 16.0 ± 6,4 | 34.2 ± 48.1 | 10.5 ± 5.6 |
| CO | Cupre | 1.6 ± 0.8 | 10.4 ± 6.5 | 47.4 ± 50.6 | 22.0 ± 0.0 | Platy | 1.6 ± 0.9 | 10.1 ± 4.9 | 34.2 ± 48.1 | 21.5 ± 1.2 |
| HC | Cupre | 2.9 ± 1.0 | 12.8 ± 6.7 | 81.6 ± 39.3 | 18.5 ± 4.6 | Platy | 3.1 ± 1.0 | 9.7 ± 5.5 | 84.2 ± 37.0 | 21.0 ± 2.4 |
| HK | Cupre | 2.9 ± 1.0 | 13.2 ± 8.2 | 89.5 ± 31.1 | 21.8 ± 0.4 | Platy | 3.1 ± 1.0 | 9.0 ± 6.2 | 81.6 ± 39.3 | 22.0 ± 0.0 |
| HM | Cupre | 2.9 ± 1.0 | 11.5 ± 5.0 | 86.8 ± 34.3 | 22.0 ± 0.0 | Platy | 3.1 ± 1.1 | 11.3 ± 4.7 | 86.8 ± 34.3 | 21.3 ± 1.0 |
| HP | Cupre | 2.7 ± 1.0 | 9.9 ± 5.0 | 84.2 ± 37.0 | 20.0 ± 3.2 | Platy | 3.3 ± 0.9 | 9.7 ± 3.8 | 86.8 ± 34.3 | 20.7 ± 2.4 |
| HS | Cupre | 2.9 ± 0.9 | 11.2 ± 6.0 | 89.5 ± 31.1 | 20.9 ± 1.2 | Platy | 3.0 ± 1.1 | 9.4 ± 5.0 | 81.6 ± 39.3 | 20.8 ± 1.8 |
| OM | Cupre | 3.1 ± 1.0 | 15.5 ± 9.5 | 89.5 ± 31.1 | 21.3 ± 1.0 | Platy | 3.3 ± 0.9 | 12.8 ± 8.4 | 94.7 ± 22.6 | 20.0 ± 3-3 |
| CO | Elaea | 1.4 ± 0.6 | 12.4 ± 5.5 | 36.8 ± 48.9 | 21.7 ± 0.8 | Pnig | 1.4 ± 0.6 | 11.1 ± 6.0 | 34.2 ± 48.1 | 21.2 ± 1.3 |
| HC | Elaea | 2.9 ± 1.0 | 17.7 ± 14.2 | 86.8 ± 34.3 | 19.6 ± 1.4 | Pnig | 2.9 ± 1.1 | 13 ± 9.0 | 14.0 ± 10.3 | 21.6 ± 3.2 |
| HK | Elaea | 3.0 ± 0.9 | 21.6 ± 18.5 | 89.5 ± 31.1 | 21.7 ± 0.8 | Pnig | 2.9 ± 1.1 | 13 ± 9.0 | 81.6 ± 39.3 | 15.0 ± 2.1 |
| HM | Elaea | 3.1 ± 0.9 | 16.6 ± 10.0 | 92.1 ± 27.3 | 21.4 ± 0.9 | Pnig | 3.2 ± 1.0 | 14.5 ± 8.8 | 86.8 ± 34.3 | 16.4 ± 5.4 |
| HP | Elaea | 2.9 ± 0.9 | 23.4 ± 17.4 | 92.1 ± 27.3 | 20.5 ± 2.1 | Pnig | 3.0 ± 1.1 | 13.4 ± 10 | 81.6 ± 39.3 | 18.6 ± 3.4 |
| HS | Elaea | 3.0 ± 0.9 | 16.5 ± 12.0 | 94.7 ± 22.6 | 20.7 ± 2.8 | Pnig | 3.4 ± 0.9 | 14.7 ± 9.7 | 92.1 ± 27.3 | 16.1 ± 5.2 |
| OM | Elaea | 2.9 ± 1.0 | 16.8 ± 11.3 | 81.6 ± 39.3 | 21.1 ± 0.8 | Pnig | 3.5 ± 0.7 | 14.5 ± 9.2 | 97.4 ± 16.2 | 12.3 ± 6.4 |
| CO | Eryt | 1.9 ± 0.9 | 22.8 ± 17.7 | 55.3 ± 50.4 | 20.8 ± 2.9 | Popul | 1.2 ± 0.5 | 31.4 ± 49.7 | 13.2 ± 34.3 | 14.1 ± 7.2 |
| HC | Eryt | 3.3 ± 1.0 | 24.9 ± 12.6 | 97.4 ± 16.2 | 21.7 ± 0.8 | Popul | 2.0 ± 1.1 | 43.4 ± 44.6 | 50.0 ± 50.7 | 19.0 ± 1.9 |
| HK | Eryt | 3.1 ± 1.0 | 25.7 ± 16.2 | 94.7 ± 22.6 | 22.0 ± 0.0 | Popul | 2.2 ± 1.1 | 33.3 ± 24.2 | 63.2 ± 48.9 | 19.0 ± 1.3 |
| HM | Eryt | 3.2 ± 1.2 | 29.4 ± 22.3 | 94.7 ± 22.6 | 21.2 ± 1.0 | Popul | 2.2 ± 1.1 | 31.3 ± 31.2 | 65.8 ± 48.1 | 13.5 ± 7.2 |
| HP | Eryt | 3.2 ± 1.0 | 23.9 ± 13.7 | 97.4 ± 16.2 | 21.3 ± 1.6 | Popul | 2.4 ± 1.1 | 34.7 ± 31.9 | 73.7 ± 44.6 | 15.0 ± 7.8 |
| HS | Eryt | 2.9 ± 1.1 | 26.3 ± 16.5 | 97.4 ± 16.2 | 21.7 ± 0.8 | Popul | 2.1 ± 1.1 | 32.2 ± 22.0 | 55.3 ± 50.4 | 16.9 ± 1.1 |

**Table A4.** *Cont.*

| Treatment | Species | Health | Height Increase (cm) | Survival (%) | Osmotic Potential (bar) | Species | Health | Height Increase (cm) | Survival (%) | Osmotic Potential (bar) |
|---|---|---|---|---|---|---|---|---|---|---|
| OM | Eryt | 3.4 ± 1.1 | 28.0 ± 16.0 | 94.7 ± 22.6 | 22.0 ± 0.0 | Popul | 2.1 ± 1.2 | 27.7 ± 23.8 | 47.4 ± 50.6 | 12.3 ± 6.4 |
| CO | Fraxi | 1.3 ± 0.5 | 20.4 ± 13.5 | 28.9 ± 46.0 | 21.5 ± 1.2 | Robin | 12.3 ± 6.4 | 21.4 ± 16.9 | 28.9 ± 46.0 | 21.7 ± 0.5 |
| HC | Fraxi | 2.7 ± 1.1 | 18.9 ± 12.9 | 81.6 ± 39.3 | 18.2 ± 8.9 | Robin | 3.0 ± 1.0 | 25.5 ± 17.7 | 84.2 ± 37.0 | 21.5 ± 1.2 |
| HK | Fraxi | 2.9 ± 1.0 | 17.1 ± 11.9 | 89.5 ± 31.1 | 19.9 ± 2.8 | Robin | 3.0 ± 1.1 | 23.1 ± 16.2 | 84.2 ± 37.0 | 20.8 ± 1.7 |
| HM | Fraxi | 2.8 ± 1.1 | 21.2 ± 17.4 | 81.6 ± 39.3 | 21.5 ± 1.2 | Robin | 2.6 ± 1.0 | 18.1 ± 10.7 | 76.3 ± 43.1 | 19.2 ± 3.0 |
| HP | Fraxi | 2.5 ± 0.9 | 28.3 ± 26.3 | 84.2 ± 37.0 | 21.3 ± 1.6 | Robin | 3.1 ± 0.9 | 34.6 ± 37.0 | 94.7 ± 22.6 | 20.0 ± 2.6 |
| HS | Fraxi | 2.7 ± 0.9 | 22.8 ± 26.4 | 84.2 ± 37.0 | 20.7 ± 3.3 | Robin | 3.2 ± 0.7 | 21.7 ± 18.5 | 94.7 ± 22.6 | 19.0 ± 3.9 |
| HS | Fraxi | 3.1 ± 1.0 | 28.2 ± 27.1 | 89.5 ± 31.1 | 20.8 ± 1.3 | Robin | 2.8 ± 1.0 | 23.4 ± 17.0 | 89.5 ± 31.1 | 19.9 ± 1.8 |

**Table A5.** Biochemical properties of trees treated with hydrogel (HS, hydrogel and subsoil) or without hydrogel (control, CO): RWC: Relative water content, C (chl a): Chlorophyll a content, C (chl b): Chlorophyll b content, LDW: Leaf dry weight, Rubisco: Rubisco content, Nitrogen: Percentage of nitrogen, EC: Electrical conductivity, Prolin: Prolin content, ABA: Abscisic acid, TP: Total protein content, and Zn: Zinc content. Values are means ± SD.

| Species | Treatment | RWC (%) | C (chl a) (mg/g FW) | C (chl b) (mg/g FW) | LDW (mg) | Rubisco (Umg/Protein) | Nitrogen (%) | EC (µs/cm) | Prolin (µmol/g FW) | ABA (ppm) | TP (mg/g DW) | Zn (mg/kg DW) |
|---|---|---|---|---|---|---|---|---|---|---|---|---|
| *Acer negundo* | CO | 72.7 ± 3.5 | 3.97 ± 0.28 | 1.20 ± 0.08 | 0.152 ± 0.02 | 89.2 ± 4.3 | 0.184 ± 0.012 | 1712 ± 75 | 1.50 ± 0.11 | 7.83 ± 0.45 | 76.5 ± 5.2 | 25.0 ± 1.4 |
| | HS | 76.2 ± 2.3 | 4.24 ± 0.23 | 1.16 ± 0.10 | 0.191 ± 0.01 | 104.3 ± 5.7 | 0.246 ± 0.015 | 1871 ± 77 | 2.05 ± 0.16 | 8.22 ± 0.41 | 90.4 ± 6.8 | 29.5 ± 2.7 |
| *Ailanthus altissima* | CO | 41.3 ± 2.6 | 4.96 ± 0.38 | 2.80 ± 0.22 | 0.166 ± 0.01 | 195.1 ± 8.5 | 0.152 ± 0.013 | 1145 ± 68 | 2.60 ± 0.18 | 5.30 ± 0.35 | 71.5 ± 5.2 | 13.5 ± 0.9 |
| | HS | 43.1 ± 1.8 | 4.25 ± 0.31 | 2.66 ± 0.21 | 0.128 ± 0.01 | 190.6 ± 8.5 | 0.136 ± 0.010 | 1138 ± 59 | 2.68 ± 0.19 | 5.72 ± 0.39 | 62.3 ± 4.3 | 16.0 ± 0.8 |
| *Azadirachta indica* | CO | 45.3 ± 2.7 | 3.18 ± 0.23 | 1.71 ± 0.14 | 0.161 ± 0.02 | 134.7 ± 6.5 | 0.161 ± 0.008 | 1425 ± 66 | 2.68 ± 0.17 | 6.15 ± 0.49 | 145.7 ± 9.12 | 20.0 ± 1.5 |
| | HS | 41.9 ± 1.3 | 2.52 ± 0.16 | 1.53 ± 0.08 | 0.120 ± 0.01 | 130.5 ± 6.1 | 0.142 ± 0.010 | 1516 ± 68 | 2.80 ± 0.18 | 7.61 ± 0.29 | 138.2 ± 10.4 | 48.0 ± 3.4 |
| *Berberis vulgaris* | CO | 71.2 ± 3.1 | 2.91 ± 0.21 | 2.82 ± 0.13 | 0.188 ± 0.02 | 93.5 ± 4.4 | 0.120 ± 0.010 | 1760 ± 80 | 0.94 ± 0.09 | 4.95 ± 0.31 | 42.9 ± 4.2 | 13.0 ± 0.6 |
| | HS | 80.4 ± 3.3 | 3.15 ± 0.17 | 2.98 ± 0.18 | 0.194 ± 0.02 | 98.2 ± 4.9 | 0.112 ± 0.080 | 1715 ± 71 | 1.21 ± 0.08 | 5.30 ± 0.33 | 50.2 ± 5.1 | 21.0 ± 0.7 |
| *Celtis caucasica* | CO | 60.4 ± 3.1 | 3.10 ± 0.21 | 2.04 ± 0.18 | 0.227 ± 0.02 | 109.1 ± 7.2 | 0.182 ± 0.013 | 1245 ± 60 | 1.82 ± 0.06 | 5.61 ± 0.32 | 49.5 ± 2.8 | 12.3 ± 0.8 |
| | HS | 63.2 ± 2.1 | 2.34 ± 0.14 | 1.61 ± 0.12 | 0.190 ± 0.02 | 103.7 ± 5.5 | 0.187 ± 0.012 | 1260 ± 58 | 2.02 ± 0.14 | 6.18 ± 0.41 | 47.4 ± 2.6 | 14.0 ± 0.5 |
| *Cercis siliquastrum* | CO | 55.3 ± 2.7 | 2.62 ± 0.21 | 2.24 ± 0.21 | 0.172 ± 0.01 | 100.2 ± 6.2 | 0.256 ± 0.019 | 1315 ± 66 | 1.50 ± 0.10 | 2.95 ± 0.20 | 60.5 ± 5.1 | 62.0 ± 3.7 |
| | HS | 57.4 ± 1.5 | 2.78 ± 0.15 | 2.74 ± 0.17 | 0.193 ± 0.01 | 117.6 ± 6.2 | 0.292 ± 0.017 | 1217 ± 52 | 1.77 ± 0.08 | 3.35 ± 0.24 | 69.3 ± 5.1 | 54.0 ± 4.9 |
| *Cupressus arizonica* | CO | 33.7 ± 1.9 | 2.95 ± 0.20 | 2.35 ± 0.20 | 0.228 ± 0.02 | 81.5 ± 6.6 | 0.191 ± 0.016 | 1914 ± 82 | 3.60 ± 0.27 | 4.48 ± 0.39 | 80.4 ± 4.2 | 33.0 ± 1.8 |
| | HS | 30.4 ± 1.3 | 2.61 ± 0.16 | 1.81 ± 0.13 | 0.207 ± 0.02 | 77.8 ± 3.5 | 0.162 ± 0.010 | 1812 ± 72 | 3.78 ± 0.27 | 5.12 ± 0.39 | 71.5 ± 5.3 | 14.5 ± 1.9 |
| *Eleagnus angustifolia* | CO | 54.2 ± 3.6 | 3.81 ± 0.28 | 1.51 ± 0.14 | 0.241 ± 0.02 | 70.7 ± 3.6 | 0.168 ± 0.012 | 1570 ± 63 | 1.72 ± 0.11 | 3.30 ± 0.25 | 218.3 ± 19.5 | 52.0 ± 2.9 |
| | HS | 52.4 ± 1.1 | 3.52 ± 0.28 | 1.15 ± 0.08 | 0.190 ± 0.02 | 66.5 ± 3.4 | 0.177 ± 0.010 | 1690 ± 67 | 1.80 ± 0.14 | 3.46 ± 0.25 | 205.0 ± 12.5 | 51.0 ± 3.2 |

**Table A5.** *Cont.*

| Species | Treatment | RWC (%) | C (chl a) (mg/g FW) | C (chl b) (mg/g FW) | LDW (mg) | Rubisco (Umg/Protein) | Nitrogen (%) | EC (μs/cm) | Prolin (μmol/g FW) | ABA (ppm) | TP (mg/g DW) | Zn (mg/kg DW) |
|---|---|---|---|---|---|---|---|---|---|---|---|---|
| *Erythrostemon gilliesii* | CO | 74.1 ± 3.8 | 3.62 ± 0.25 | 2.12 ± 0.16 | 0.161 ± 0.01 | 81.5 ± 6.2 | 0.164 ± 0.017 | 1325 ± 72 | 2.20 ± 0.12 | 2.96 ± 0.24 | 40.4 ± 3.1 | 81.0 ± 5.9 |
| | HS | 74.7 ± 3.0 | 3.42 ± 0.21 | 1.75 ± 0.14 | 0.147 ± 0.01 | 79.1 ± 4.1 | 0.153 ± 0.010 | 1248 ± 53 | 2.63 ± 0.18 | 3.40 ± 0.26 | 36.2 ± 2.8 | 25.5 ± 1.8 |
| *Fraxinus excelsior* | CO | 50.5 ± 2.9 | 3.81 ± 0.24 | 2.91 ± 0.22 | 0.155 ± 0.01 | 113.1 ± 5.2 | 0.202 ± 0.014 | 1305 ± 58 | 2.85 ± 0.12 | 1.50 ± 0.08 | 52.3 ± 4.6 | 17.5 ± 0.8 |
| | HS | 52.7 ± 1.4 | 3.98 ± 0.27 | 3.27 ± 0.21 | 0.171 ± 0.01 | 118.6 ± 6.3 | 0.226 ± 0.014 | 1174 ± 53 | 3.10 ± 0.24 | 1.83 ± 0.11 | 58.6 ± 4.4 | 26.0 ± 1.7 |
| *Juglans nigra* | CO | 59.2 ± 3.7 | 4.72 ± 0.23 | 2.58 ± 0.18 | 0.256 ± 0.02 | 98.6 ± 4.7 | 0.268 ± 0.008 | 1815 ± 81 | 4.12 ± 0.27 | 1.46 ± 0.09 | 139.5 ± 11.4 | 36.0 ± 2.0 |
| | HS | 61.1 ± 2.8 | 5.11 ± 0.39 | 2.78 ± 0.20 | 0.296 ± 0.02 | 116.6 ± 6.9 | 0.308 ± 0.022 | 1850 ± 71 | 4.45 ± 0.37 | 1.95 ± 0.09 | 143.1 ± 10.2 | 44.0 ± 2.9 |
| *Juniperus sp.* | CO | 28.8 ± 1.3 | 4.03 ± 0.38 | 1.91 ± 0.14 | 0.252 ± 0.01 | 68.8 ± 3.5 | 0.228 ± 0.019 | 1390 ± 53 | 2.95 ± 0.18 | 3.52 ± 0.29 | 100.4 ± 9.2 | 15.0 ± 1.4 |
| | HS | 34.5 ± 1.2 | 4.55 ± 0.34 | 2.50 ± 0.16 | 0.285 ± 0.02 | 83.4 ± 4.2 | 0.255 ± 0.013 | 1288 ± 59 | 3.20 ± 0.18 | 3.80 ± 0.25 | 112.4 ± 8.2 | 15.5 ± 0.9 |
| *Morus alba* | CO | 67.8 ± 2.5 | 2.92 ± 0.21 | 1.80 ± 0.11 | 0.128 ± 0.01 | 129.4 ± 6.1 | 0.208 ± 0.014 | 1834 ± 83 | 1.12 ± 0.08 | 9.83 ± 0.71 | 37.8 ± 2.9 | 52.0 ± 2.8 |
| | HS | 74.8 ± 2.1 | 3.27 ± 0.22 | 2.11 ± 0.14 | 0.159 ± 0.01 | 144.5 ± 7.1 | 0.238 ± 0.016 | 2085 ± 97 | 1.63 ± 0.07 | 10.59 ± 0.46 | 46.8 ± 3.2 | 57.5 ± 2.6 |
| *Olea europea* | CO | 41.9 ± 3.2 | 5.21 ± 0.31 | 2.10 ± 0.18 | 0.288 ± 0.02 | 97.6 ± 5.1 | 0.181 ± 0.010 | 1810 ± 77 | 1.35 ± 0.09 | 4.77 ± 0.31 | 128.4 ± 9.4 | 27.0 ± 1.3 |
| | HS | 43.2 ± 1.3 | 4.47 ± 0.31 | 1.52 ± 0.10 | 0.240 ± 0.02 | 94.3 ± 5.1 | 0.198 ± 0.011 | 1895 ± 61 | 1.50 ± 0.09 | 5.20 ± 0.31 | 120.1 ± 8.2 | 8.0 ± 0.6 |
| *Pinus eldarica* | CO | 31.5 ± 2.0 | 4.61 ± 0.31 | 1.62 ± 0.12 | 0.241 ± 0.02 | 60.3 ± 3.8 | 0.254 ± 0.019 | 1746 ± 63 | 4.80 ± 0.28 | 2.56 ± 0.16 | 96.3 ± 7.2 | 13.5 ± 1.1 |
| | HS | 28.1 ± 0.7 | 4.41 ± 0.32 | 1.22 ± 0.10 | 0.181 ± 0.02 | 55.7 ± 2.9 | 0.263 ± 0.016 | 1672 ± 68 | 4.88 ± 0.38 | 2.70 ± 0.16 | 90.5 ± 8.1 | 30.5 ± 2.4 |
| *Pinus nigra* | CO | 38.5 ± 1.7 | 4.02 ± 0.31 | 2.37 ± 0.18 | 0.261 ± 0.02 | 109.1 ± 8.2 | 0.261 ± 0.019 | 1845 ± 71 | 5.75 ± 0.32 | 1.98 ± 0.12 | 108.2 ± 9.4 | 10.5 ± 0.9 |
| | HS | 34.2 ± 1.1 | 4.18 ± 0.31 | 2.41 ± 0.20 | 0.248 ± 0.02 | 151.5 ± 7.2 | 0.250 ± 0.017 | 1738 ± 60 | 5.78 ± 0.36 | 2.40 ± 0.12 | 111.5 ± 8.4 | 34.5 ± 2.9 |
| *Platanus orientalis* | CO | 72.4 ± 3.7 | 3.72 ± 0.29 | 1.86 ± 0.14 | 0.160 ± 0.01 | 169.8 ± 7.5 | 0.216 ± 0.014 | 1220 ± 62 | 1.92 ± 0.15 | 5.42 ± 0.37 | 62.5 ± 4.3 | 22.0 ± 1.2 |
| | HS | 78.8 ± 2.4 | 4.08 ± 0.21 | 2.26 ± 0.17 | 0.188 ± 0.01 | 182.0 ± 8.4 | 0.263 ± 0.018 | 1214 ± 52 | 1.88 ± 0.11 | 5.05 ± 0.38 | 77.3 ± 5.6 | 24.0 ± 1.6 |
| *Platycladus orientalis* | CO | 36.4 ± 1.4 | 3.93 ± 0.28 | 1.81 ± 0.13 | 0.258 ± 0.02 | 41.4 ± 3.6 | 0.215 ± 0.014 | 1695 ± 75 | 3.86 ± 0.22 | 4.91 ± 0.38 | 120.3 ± 8.8 | 11.0 ± 0.7 |
| | HS | 30.5 ± 1.4 | 3.72 ± 0.21 | 1.34 ± 0.12 | 0.208 ± 0.02 | 38.1 ± 4.0 | 0.203 ± 0.014 | 1569 ± 59 | 4.12 ± 0.35 | 5.21 ± 0.37 | 104.3 ± 7.3 | 20.0 ± 1.9 |
| *Populus nigra* | CO | 71.5 ± 3.7 | 4.10 ± 0.37 | 1.93 ± 0.17 | 0.182 ± 0.02 | 98.5 ± 4.6 | 0.236 ± 0.019 | 1316 ± 75 | 2.26 ± 0.18 | 1.95 ± 0.16 | 41.2 ± 3.6 | 65.0 ± 6.9 |
| | HS | 76.2 ± 2.2 | 3.77 ± 0.26 | 1.82 ± 0.13 | 0.146 ± 0.01 | 96.3 ± 4.4 | 0.221 ± 0.017 | 1171 ± 66 | 2.42 ± 0.19 | 2.30 ± 0.09 | 37.5 ± 2.8 | 182.5 ± 8.7 |
| *Robinia pseudoacacia* | CO | 59.6 ± 2.7 | 3.91 ± 0.29 | 3.22 ± 0.19 | 0.195 ± 0.01 | 112.2 ± 5.4 | 0.242 ± 0.019 | 2190 ± 88 | 0.93 ± 0.07 | 2.65 ± 0.19 | 76.3 ± 5.3 | 32.0 ± 3.9 |
| | HS | 55.4 ± 2.2 | 4.28 ± 0.26 | 3.42 ± 0.26 | 0.254 ± 0.01 | 120.3 ± 6.2 | 0.277 ± 0.018 | 2307 ± 84 | 1.15 ± 0.09 | 3.25 ± 0.19 | 92.8 ± 7.4 | 42.5 ± 3.6 |

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
