# Peer review of "Effect of Different Soil Treatments with Hydrogel on the Performance of Drought-Sensitive and Tolerant Tree Species in a Semi-Arid Region"

_forests, doi:10.3390/f11020211_

Round 1

Reviewer 1 Report

This is an interesting study that investigates the effects of hydrogel application along with various materials on the performance of 20 species. The study found generally positive responses of plant performance, especially for drought-sensitive species.

While the study conducted detailed measurements and analyses, I concern about the ways to deliver and visualize them in the manuscript. The study included 20 species and 7 treatments, including control. Because of this large number of combinations, the current way of displaying the results (e.g. X1, X2 and X3 treatments increased a trait Y of Z1, Z2 and Z3 species) doesn’t seem effective to deliver the main idea of this study to readers. Furthermore, full tables for individual species and treatments are not helpful for readers to check and to evaluate key findings in the main text. I strongly suggest to make groups for species (e.g. broadleaved species and conifers) or treatment (e.g. hydrogel + compost across all species) according to the main idea to be delivered in each paragraph, and to visualize as figures (e.g. boxplot or scatter plot with error bars). Full tables can be moved to the appendix section.  

This manuscript combined results and discussion sections as one. The section covered results of this study generally well, but discussions on the results seemed not enough. Although it is authors’ choice how to organize the paper, I would like to suggest to separate the two sections. By separating discussion section from results, readers may understand key findings of the study and their implications better.

Both table 3 and table 4 introduced biochemical properties from different species and treatment combinations. Unless there is a specific reason to separate the two groups of biochemical properties, it should be one table. Again, I suggest to move both tables to appendix section, and to consider figures to show results.

In tables 3 and 4, all hydrogel treatments were combined and compared to control treatment. Authors need to show that differences among the treatments are not significant. Or, there should be a reasoning for comparison as current way.       

Minor comments:

L64: Different planting approaches?

L112-117: Please add equation numbers and describe abbreviations as a sentence.

L109-136: For simple descriptions of the biochemical analyses, a table would be better than a series of sentences after colons.

L136: Please add an equation number.

L138-139: Citation for R program?

L144-145: Is it possible to make a figure for the increase in height instead of the table?  

L193-196: Confirmed? Results of this study is inconsistent with one previous study. It is not confirmed, and potential reason for the inconsistent result need to be discussed.

L199: “by increasing root exudates” - references for increase in root exudates with hydrogel application need to be cited.

Author Response

Response to Reviewer 1 Comments

Point 1: This is an interesting study that investigates the effects of hydrogel application along with various materials on the performance of 20 species. The study found generally positive responses of plant performance, especially for drought-sensitive species.

While the study conducted detailed measurements and analyses, I concern about the ways to deliver and visualize them in the manuscript. The study included 20 species and 7 treatments, including control. Because of this large number of combinations, the current way of displaying the results (e.g. X1, X2 and X3 treatments increased a trait Y of Z1, Z2 and Z3 species) doesn’t seem effective to deliver the main idea of this study to readers. Furthermore, full tables for individual species and treatments are not helpful for readers to check and to evaluate key findings in the main text. I strongly suggest to make groups for species (e.g. broadleaved species and conifers) or treatment (e.g. hydrogel + compost across all species) according to the main idea to be delivered in each paragraph, and to visualize as figures (e.g. boxplot or scatter plot with error bars). Full tables can be moved to the appendix section. 

Response 1: We agree. We divided all the tree species and treatments according to following rules: drought tolerant and drought sensitive with respect to P50 value and differences in biochemical parameters. Cupressus, Olea, Pinus sp. were part of the first group (abbreviation: TOL = drought tolerant). The rest of the species were part of the second group (abbreviation: SEN = drought sensitive). All treatments with hydrogel were matched into one hydrogel group (abbreviation: HYDRO = hydrogel treatment) and compared to the control one (abbreviation CO). The box plots were displayed for the growth increase. Healthy and survival status were displayed in form of column plots because of their variable type (discrete values). We implemented the key findings into the text. The tables for all treatments and tree species separately were moved into appendix section.

Point 2: This manuscript combined results and discussion sections as one. The section covered results of this study generally well, but discussions on the results seemed not enough. Although it is authors’ choice how to organize the paper, I would like to suggest to separate the two sections. By separating discussion section from results, readers may understand key findings of the study and their implications better.

Response 2: The Results section was separated from discussion for deeper discussion of our obtained results.

Point 3: Both table 3 and table 4 introduced biochemical properties from different species and treatment combinations. Unless there is a specific reason to separate the two groups of biochemical properties, it should be one table. Again, I suggest to move both tables to appendix section, and to consider figures to show results.

Response 3: Similarly, to the reorganizing the groups for morphological and vitality parameters (health, height increase a survival) we created two groups for treatments (HS and CO) and for tree species (SEN and TOL), respectively.

We put a mistake in the headings of the table as the Tr. abbreviation is a treatment with subsoil (e.g. HS) and not the average of all treatments with hydrogel addition.

Spider plot was created for better comparison of investigated parameters (for better visibility without deviations). For each plant group, values are displayed relative to CO treatment and each concentric segment of a web denotes 10% difference in given parameter. Nevertheless, the table with means and standard deviations is only one and is part of the appendix (Table A5).

Point 4: In tables 3 and 4, all hydrogel treatments were combined and compared to control treatment. Authors need to show that differences among the treatments are not significant. Or, there should be a reasoning for comparison as current way. 

Response 4: There were two reasons not to compare all treatments among them on all tree species. The first one was the financial reason because many of the laboratory work are too expensive. The second reason was that did not expect statistically significant differences among the treatments (e.g. treatment with hydrogel and subsoil and treatment with hydrogel and wheat straw).     

Minor comments:

L64: Different planting approaches?

Answer: Corrected to “different mixture treatments with hydrogel content” Lines 64-65.

L112-117: Please add equation numbers and describe abbreviations as a sentence.

Answer: Done. Lines:110-111.

L109-136: For simple descriptions of the biochemical analyses, a table would be better than a series of sentences after colons.

Answer: Done. Lines:110-111.

L136: Please add an equation number.

Answer: Done. Lines:110-111.

L138-139: Citation for R program?

Answer: Statistical analysis was conducted using R software environment, version 3.6.1 (R Core Team, 2019), and plot was made using package fmsb. Analysis of variance (one way ANOVA) and to the Tukey’s HSD post-hoc test were used to reveal the differences among investigated treatments and groups of tree species. Lines:121-124.

L144-145: Is it possible to make a figure for the increase in height instead of the table? 

Answer: Done. Figure 1.

L193-196: Confirmed? Results of this study is inconsistent with one previous study. It is not confirmed, and potential reason for the inconsistent result need to be discussed.

Answer: Yes, it was not correct in our manuscript. The sentence was overwrite: Treatments with hydrogel content are not suitable for any tree species and their different susceptibility to environmental stress is a reason of conflicting results. Lines:189-192.

L199: “by increasing root exudates” - references for increase in root exudates with hydrogel application need to be cited.

Answer: It was a speculation and thus the sentence was removed.

Reviewer 2 Report

The title of the paper is “Hydrogel treatment…” There is no real comparison between control and hydrogel alone. All the treatment included other substance such as sawdust, compost etc, therefore it was not clear whether the difference was coming from hydrogel or other materials.

In addition, there was no statistical comparison in Table 3 and 4. The authors only provided number without statistics.

Pg 2. Ln36: Although Iran is mostly an arid or semi-arid country, climate change is causing that part of the country with a temperate climate near the Caspian Sea to become semi-arid. >> Sentence is odd. Remove “that” to make a simple sentence or remove “to” to make a complex sentence with a clause starting with “that part of the country…..”.

Pg 3. Ln63: positive effects on plant growth of hydrogel >>> positive effects of hydrogel on plant growth

Pg 3. Ln70: The research was conducted in 2008 in an area 15 km east of Tehran >>> The research was conducted at 15 km east of Tehran in 2008.

Pg 3. Ln71: The average annual precipitation was 208 mm, and most precipitation occurred in the winter. >>> Provide reference and the variation around the average.

Pg 3. Ln79: There is no hydrogel alone treatment, therefore the effect of treatment cannot be separated into hydrogen and other materials.

Pg 3. Ln94: The fertiliser in form of capsules (10 g, with N, P, K nutrients) was put to the roots together with the hydrogel. >>> Was the fertilizer added to the control plot? If so, please describe it.

Pg 3. Ln104: One table should be located on the same page. In addition, give the species specific reference for P50.

Pg 7. Ln162: Please provide the mean with variation in Table 2. In other words, mean ±s.e, just like Table 3.

Table 3 and4: There are no statistical comparison. They have mean and variation, but no evidence whether there are statistical differences between treatments or not. Therefore, I was not sure whether there was difference or not.

Author Response

Response to Reviewer 2 Comments

Point 1: The title of the paper is “Hydrogel treatment…” There is no real comparison between control and hydrogel alone. All the treatment included other substance such as sawdust, compost etc, therefore it was not clear whether the difference was coming from hydrogel or other materials.

In addition, there was no statistical comparison in Table 3 and 4. The authors only provided number without statistics.

Response 1: Yes, we changed the title of the manuscript with respect to the treatments. Statistical comparison of all treatments among all tree species are part of the Supplementary material (Appendix).

For better visibility the groups all the tree species and treatments were divided according to following rules: drought tolerant and drought sensitive with respect to P50 value and differences in biochemical parameters. Cupressus, Olea, Pinus sp. were part of the first group (abbreviation: TOL = drought tolerant). The rest of the species were part of the second group (abbreviation: SEN = drought sensitive). All treatments with hydrogel were matched into one hydrogel group (abbreviation: HYDRO = hydrogel treatment) and compared to the control one (abbreviation CO).

Point 2: Pg 2. Ln36: Although Iran is mostly an arid or semi-arid country, climate change is causing that part of the country with a temperate climate near the Caspian Sea to become semi-arid. >> Sentence is odd. Remove “that” to make a simple sentence or remove “to” to make a complex sentence with a clause starting with “that part of the country…..”.

Response 2: Done. Part of the country with a temperate climate near the Caspian Sea is becoming semi-arid because of the climate change. Lines 36-38.

Point 3: Pg 3. Ln63: positive effects on plant growth of hydrogel >>> positive effects of hydrogel on plant growth

Response 3: Done. Lines 64-65.

Point 4: Pg 3. Ln70: The research was conducted in 2008 in an area 15 km east of Tehran >>> The research was conducted at 15 km east of Tehran in 2008.

Response 4: Done. Lines 71-72.     

Point 5: Pg 3. Ln71: The average annual precipitation was 208 mm, and most precipitation occurred in the winter. >>> Provide reference and the variation around the average.

Response 5: Mean annual rainfall ranged from 51 mm to 1835 mm (Javari, 2016). Lines 72-74.     

Point 6: Pg 3. Ln79: There is no hydrogel alone treatment, therefore the effect of treatment cannot be separated into hydrogen and other materials.

Response 6: Yes, we analyzed either all treatments with hydrogel addition (part of the appendix). Or we made groups based on comparison control and all other treatments with hydrogel portion for better visibility if the obtained results.

Point 7: Pg 3. Ln94: The fertiliser in form of capsules (10 g, with N, P, K nutrients) was put to the roots together with the hydrogel. >>> Was the fertilizer added to the control plot? If so, please describe it.

Response 7: No, there was no fertilizer at the control plot.

Point 8: Pg 3. Ln104: One table should be located on the same page. In addition, give the species specific reference for P50.

Response 8: References were added to the Table 1.

Point 9: Pg 7. Ln162: Please provide the mean with variation in Table 2. In other words, mean ±s.e, just like Table 3.

Response 9: Variation was added. See Table A4.

Point 10: Table 3 and4: There are no statistical comparison. They have mean and variation, but no evidence whether there are statistical differences between treatments or not. Therefore, I was not sure whether there was difference or not.

Response 10: The statistical comparison between two groups (SEN and TOL) was done. The differences are mentioned in the text Figure 1-3.

Round 2

Reviewer 1 Report

The manuscript seemed to have more solid story after the revision. I have some minor comments on the manuscript.

Minor comments:

L108: Equation 3 may have some typos (missing c for car and chl b; capitalized c for chl a). The First line below equation 3 is also not clean (possibly due to typos). What is ch a/b?

L109-113: I believe that the authors have tested the effects of different hydrogel treatments, but presented the results as groups in the main figures (with detailed results in the appendix). This text can be misread that all treatments were combined before all the tests, which is not consistent with the experimental setting.

L142-146: Both figure 3 and table A5 do not have HM treatment. Maybe typos?

Figure 3: In the previous two figures, SEN was on the left and TOL was on the right. It confused me a bit.

L174: Comma in a wrong place.

Author Response

Minor comments:

L108: Equation 3 may have some typos (missing c for car and chl b; capitalized c for chl a). The First line below equation 3 is also not clean (possibly due to typos). What is ch a/b?

Answer: Yes, the equations and description could be confusing. The part was corrected.

Concentration of chlorophyll a [c(chl a)] and b [c(chl b)] and carotenoids c(car) is calculated from the pigment absorbance at different wavelengths (A663, 646, 470) released by spectrophotometer.

L109-113: I believe that the authors have tested the effects of different hydrogel treatments, but presented the results as groups in the main figures (with detailed results in the appendix). This text can be misread that all treatments were combined before all the tests, which is not consistent with the experimental setting.

Answer:

In order to make the statistical results clearer the findings were expressed for two treatments only: control one and all treatments with hydrogel addition were embodied into one hydrogel treatment (although no pure hydrogel treatment was used). All treatments with hydrogel addition (e.g. OM, HM, HC, HK, HS, HP) were joined together into one group. Finally, two groups were compared – control and hydrogel treatments. Nevertheless, the detailed results of statistical differences among all tree species and all seven treatments are part of the appendix.

L142-146: Both figure 3 and table A5 do not have HM treatment. Maybe typos?

Answer: No, it is correct. We mentioned in the methodology that biochemical analyses were done for control and HS treatments only (hydrogel with subsoil).

Figure 3: In the previous two figures, SEN was on the left and TOL was on the right. It confused me a bit.

Answer: Yes, we switched it to the original version. SEN is on the left and TOL on the right.

L174: Comma in a wrong place.

Answer: Yes, the mistake was fixed. This corrected sentence was inserted into the text:

Similarly to results of M´barki [29] hydrogel increased growth, dry weight, RWC because of stronger uptake of the micro- and macronutrients, especially nitrogen and potassium [30].

Reviewer 2 Report

This version of paper has been improved a lot and resolved most of issues raised last time.

Pg 2. Ln36: Although Iran is mostly an arid or semi-arid country. Part of the country with a temperate climate near the Caspian Sea is becoming semi-arid because of the climate change >> Two sentences are better to be one sentence. Although Iran is mostly an arid or semi-arid country, part of the country with a temperate climate near the Caspian Sea is becoming semi-arid because of the climate change.

Pg 3. Ln71: The average annual precipitation was 208 mm, and most precipitation occurred in the winter. >>> Provide reference and the variation around the average. Even though the authors claimed they have added “Mean annual rainfall ranged from 51 mm to 1835 mm (Javari, 2016). Lines 72-74.” It was not found in the revised manuscript.

Pg 3. Ln94: The fertiliser in form of capsules (10 g, with N, P, K nutrients) was put to the roots together with the hydrogel. >>>The authors should clarify there was no fertilizer at the control plots in the manuscript.

Pg 4. The activity of ribulose-l,5-bisphosphate carboxylase oxygenase (Rubisco) was measured with a spectrophotometer according to the [24]. >>> according to Barta et al. 2011 {24}

Pg 4. Ln109: In order to make the statistical results clearer >>> In order to make the statistical results clear.

Author Response

Review 2

This version of paper has been improved a lot and resolved most of issues raised last time.

Pg 2. Ln36: Although Iran is mostly an arid or semi-arid country. Part of the country with a temperate climate near the Caspian Sea is becoming semi-arid because of the climate change >> Two sentences are better to be one sentence. Although Iran is mostly an arid or semi-arid country, part of the country with a temperate climate near the Caspian Sea is becoming semi-arid because of the climate change.

Answer: Yes, corrected as suggested: Although Iran is mostly an arid or semi-arid country, part of the country with a temperate climate near the Caspian Sea is becoming semi-arid because of the climate change.

Pg 3. Ln71: The average annual precipitation was 208 mm, and most precipitation occurred in the winter. >>> Provide reference and the variation around the average. Even though the authors claimed they have added “Mean annual rainfall ranged from 51 mm to 1835 mm (Javari, 2016). Lines 72-74.” It was not found in the revised manuscript.

Answer: Yes, we omitted to put the sentence into the manuscript. Corrected: Mean annual rainfall ranged from 51 mm to 1835 mm [17].

Pg 3. Ln94: The fertiliser in form of capsules (10 g, with N, P, K nutrients) was put to the roots together with the hydrogel. >>>The authors should clarify there was no fertilizer at the control plots in the manuscript.

Answer: Yes, this sentence was added into the methodology section: There was no fertilizer at the control plots.

Pg 4. The activity of ribulose-l,5-bisphosphate carboxylase oxygenase (Rubisco) was measured with a spectrophotometer according to the [24]. >>> according to Barta et al. 2011 {24}

Answer: Corrected.
